# *Bacillus cereus* in the Artisanal Cheese Production Chain in Southwestern Mexico

**DOI:** 10.3390/microorganisms11051290

**Published:** 2023-05-15

**Authors:** Itzel-Maralhi Cruz-Facundo, Jeiry Toribio-Jiménez, Natividad Castro-Alarcón, Marco-Antonio Leyva-Vázquez, Hugo-Alberto Rodríguez-Ruíz, José-Humberto Pérez-Olais, Roberto Adame-Gómez, Elvia Rodríguez-Bataz, Joel Reyes-Roldán, Salvador Muñoz-Barrios, Arturo Ramírez-Peralta

**Affiliations:** 1Laboratorio de Investigación en Patometabolismo Microbiano, Universidad Autónoma de Guerrero, Guerrero 39074, Mexico; maralhi.09@gmail.com (I.-M.C.-F.); robert94a25@gmail.com (R.A.-G.);; 2Laboratorio de Investigación en Microbiología Molecular y Biotecnología Ambiental, Universidad Autónoma de Guerrero, Guerrero 39074, Mexico; jeiryjimenez2014@gmail.com; 3Laboratorio de Investigación en Microbiologia, Universidad Autónoma de Guerrero, Guerrero 39074, Mexico; natividadcastro24@gmail.com; 4Laboratorio de Investigación en Biomedicina Molecular, Universidad Autónoma de Guerrero, Guerrero 39074, Mexico; leyvamarco13@gmail.com (M.-A.L.-V.); hugordzrz@gmail.com (H.-A.R.-R.); 5Laboratorio de Investigación en Obesidad y Diabetes, Universidad Autónoma de Guerrero, Guerrero 39074, Mexico; 6Unidad de Investigación en Virología y Cancer, Hospital Infantil de México Federico Gomez, Ciudad de Mexico 06720, Mexico; jholais@gmail.com; 7Laboratorio de Investigación en Parasitologia, Universidad Autónoma de Guerrero, Guerrero 39074, Mexico; 8Laboratorio de Investigación en Inmunotoxigenomica, Universidad Autónoma de Guerrero, Guerrero 39074, Mexico; smunoz@uagro.mx

**Keywords:** *Bacillus cereus*, cheeses, Mexico, enterotoxins

## Abstract

Background: *Bacillus cereus* is associated with milk, dairy product, and dairy farm contamination. The aim of this study was to characterize strains of *B. cereus* in the small-scale artisanal cheese production chain in southwestern Mexico. Methods: 130 samples were collected. *B. cereus* isolation was performed on Mannitol Egg Yolk Polymyxin (MYP) agar. Genotyping, enterotoxigenic profile, and determination of genes involved in the formation of *B. cereus* biofilm were performed by PCR. An antimicrobial susceptibility test was made by broth microdilution assay. The phylogenetic analysis was performed by amplification and sequencing of 16s rRNA. Results: *B. cereus sensu lato* was isolated and molecularly identified in 16 samples and *B. cereus sensu stricto* (*B. cereus*) was the most frequently isolated and identified species (81.25%). Of all the isolated *B. cereus sensu lato* strains, 93.75% presented at least one gene for some diarrheagenic toxins, 87.5% formed biofilms, and 18.75% were amylolytic. All *B. cereus sensu lato* strains were resistant to beta-lactams and folate inhibitors. A close phylogenetic relationship between isolates was found between the cheese isolates and the air isolates. Conclusions: Strains of *B. cereus sensu lato* were found in small-scale artisanal cheeses on a farm in southwestern Mexico.

## 1. Introduction

*Bacillus cereus* is a rod-shaped, Gram-positive, endospore-forming, facultative anaerobic bacterium associated with food poisoning worldwide [1,2]. *B. cereus* shares a high phenotypic and genotypic similarity with other *Bacillus* species, so they have been included in the *B. cereus* group or *B. cereus sensu lato* (*s.l.*) [3,4]. *B. cereus s.l.* is ubiquitous in the environment, particularly in different types of soil, sediment, dust, and plants, due to the resistance of the spores to adverse environmental conditions [5,6]. *B. cereus sensu stricto* (*B. cereus*) is an opportunistic pathogen that causes two types of intoxication in humans from consuming contaminated food: the emetic syndrome, which causes nausea and vomiting; and the diarrheal syndrome, characterized by abdominal pain and diarrhea [7]. The pathogenicity of *B. cereus* is attributed to the production of different toxins. The emetic syndrome is caused by ingesting the emetic toxin, which is pre-formed in food during growth by *B. cereus* [8,9]. This toxin, called cereulide, is stable to heat, extreme pH conditions, and protease activity [10]. The cereulide is synthesized by a non-ribosomal peptide synthetase encoded in the *ces* operon [11]. The diarrheal syndrome is associated with the production of enterotoxins such as hemolysin BL (Hbl), nonhemolytic enterotoxin (Nhe), and cytotoxin K (CytK) [12,13,14,15]. In addition, enterotoxin FM (EntFM), enterotoxin T (becT), and enterotoxin S (EntS) have been described as potential diarrheal toxins; however, the possible role of these in the development of the disease is not clear [16,17,18].

This microorganism has been isolated from diverse foods, including red meat, rice, pasta, condiments, raw fruits, spices, vegetables, ready-to-eat foods, and dairy products [5]. The monitoring of *B. cereus* in milk and dairy products is important since this microorganism is also a food spoilage microorganism that can negatively affect the quality of the product, reduce its nutritional value, and shorten its shelf life. In this sense, *B. cereus* produces heat-stable extracellular enzymes such as proteases and lipases [19,20]. Proteases can cause a sour taste and gelatinization of milk, while lipases are associated with a rancid taste [21]. The heat-stable spores of *B. cereus* in milk are a source of contamination for milk-derived products, as the spores are resistant to milk pasteurization processes. Therefore, the isolation of *B. cereus* in pasteurized milk is not surprising [22,23]. In this sense, it is essential to consider the isolates in raw milk [24], as many of these dairy products are still made with this type of milk [25,26,27]. In addition to spore production, *B. cereus* has been described to produce biofilms in food production environments, and this process has been considered as a possible way that products are systematically contaminated [28,29]. *B. cereus* has been isolated not only from dairy products but also from dairy farms, mainly from cattle feces, milk tanks, and feedlot feed [30,31,32]; which can explain its presence in the final product.

Queso Fresco (QF) is a fresh Hispanic-style cheese that is soft, fresh, unpressed, and unripened, obtained by raw milk enzymatic coagulation [26]. QF, Adobera, Oaxaca, Asadero, and Panela are commonly consumed in Mexico [33]. However, the presence of pathogens associated with food poisoning, such as *Listeria monocytogenes*, *Staphylococcus aureus*, *Salmonella*, and *B. cereus* has been reported in these products [25,26,34,35,36]. In the case of *B. cereus* in dairy products in Mexico, there is only one previous report on artisanal dairy products sold in the southwest of the country, which reports a high frequency of *B. cereus s.l.* in QF [25]. Even though there are previous studies of *B. cereus* in commercialized cheeses, it is crucial to evaluate *B. cereus* from the dairy farm, during production, and until the product’s sale. Therefore, the aim of this study was to isolate and characterize strains of *B. cereus s.l.* in the small-scale QF production chain in southwestern Mexico.

## 2. Materials and Methods

### 2.1. Sample Collection

In August 2021, 130 environmental samples, including 9 one-kilogram samples of cheese, were collected from a dairy farm in southwestern Mexico. Of the dairy farm environmental samples, 9 were from silty soil, 30 from milk, 15 from the air, 18 from buckets of milk, 27 from the surfaces of farm animals, 9 from pets, 8 from cow feces, and 5 from workers’ nostrils. All the samples were obtained throughout the day, starting at 4:00 a.m. with the obtaining of the milk until the pressing and obtaining of the cheese. The 9 kg of cheese represented the production of one day.

Soil samples were collected in sterile jars from different locations on the farm. Milk samples were collected by qualified personnel by milking the cow. Air samples were collected using the sinking method by exposing Mannitol Egg Yolk Polymyxin (MYP) agar plates to the air for 10 min at different points of the farm. Skin samples from domestic and farm animals and milk containers were collected using the method of covering the surface, which consists of sliding sterile swabs immersed in saline solution over a 5 cm plane; these swabs were preserved in Stuart transport medium until analysis. Bovine feces were collected in sterile containers using wooden applicators. Nostril samples were collected from the anterior nostrils of the milkers with a sterile swab.

### 2.2. Isolation and Identification of Bacterial Strains

The swabs and 100 µL of milk samples were inoculated directly onto MYP agar plates. In the case of cheese samples, soil and cow feces were processed as solid foods according to NOM-110-SSA1-1994 [37]. The MYP plates of all the samples analyzed, including those exposed to the environment, were incubated at 30 °C for 24 h. Pink colonies with a precipitation halo were selected as being suspicious for *B. cereus* [38]. These strains were cultured in Brain Heart Infusion (BHI) broth for storage and subsequent analysis.

### 2.3. Genotyping and Enterotoxigenic Profile of B. cereus

For DNA extraction, 1 mL of a 24 h liquid bacterial culture was centrifuged for 10 min at 10,000 rpm. The pellet was resuspended in 200 μL of lysis solution (10 mM Tris-HCl, 1 mM EDTA pH 8.0, and 1 mg/mL lysozyme) and incubated for 30 min at 37 °C. Subsequently, 250 μL of phenol-chloroform, isoamyl alcohol (ratio 25:24:1), was added and homogenized by inversion. It was centrifuged at 10,000 rpm for 5 min, and 200 μL of the aqueous phase was recovered, mixed with 1 mL of cold absolute ethanol, and centrifuged at 10,000 rpm for 10 min. The supernatant was completely removed, and the DNA was resuspended in 20 μL of TE buffer. The molecular identification of the strains of *B. cereus s.l.* was made from the amplification of the *16s rRNA* gene (*16s rRNA* F- AGAGTTTGATCCTGGCTC, *16s rRNA* R-CGGCTACCTTGTTACGAC) [39], performing end-point PCR with the following reaction mixture: 0.2 mM of each dNTP, 3 mM MgCl2, 0.2 μM of each oligonucleotide, 1 U of Taq DNA polymerase (Ampliqon^®^, DEN), 1X Buffer, and 100 ng of DNA. The PCR protocol was: one cycle at 95 °C for 5 min, 35 cycles at 95 °C for 30 s, 55 °C for 30 s, 72 °C for 90 s, and one cycle at 72 °C for 10 min. For the determination of the phylogenetic relationship between the strains of *B. cereus s.l.*, the purification of the PCR products was performed with the FastGene Gel/PCR Extraction Kit for the strains that amplified the *16s rRNA* gene; subsequently, this was sent for sequencing to the Institute of Biotechnology of the National Autonomous University of Mexico. The identification of the species of *B. cereus s.l.* was performed in EzBioCloud platform from the sequence analysis.

Detection of *B. cereus* toxin genes was performed by PCR of conserved regions of the operons *nhe*ABC (*nhe*-F GCCCTGGTATGTATATTGGATCTAC, *nhe*-R GGTCATAATATCTTCTACAGCAAGG) and *hbl*ACD (*hbl*-F GTAAATTAIGATGAICAATTTC, *hbl*-R AGAATAGGCATTGATAGATT), which code for the non-hemolytic enterotoxins and BL hemolysin, respectively; and the genes *ces* (*ceS*-F GGTGACACATTATCATATAAGGTG, *ceS*-R GTAAGCGAACCTGTCTGTAACAACA) and *cytK* (*cytK*-F CAAAACTCATCTATGCAATTATGCAT, *cytK*-R ACCAGTTGTATTAATAACGGCAATC), which code for the emetic toxin and cytotoxin K, respectively. For each PCR, the mix contained the following: 25 μL of REDTaq DNA Polymerase Ready Mix (Sigma-Aldrich, St. Louis, MO, USA), 11 μL of sterile MiliQ water, 10 to 20 ng of genomic DNA, and 0.02 μM of each oligonucleotide. The conditions used for the *nhe*, *hbl*, and *ces* genes were 1 cycle at 94 °C for 5 min, 35 cycles at 94 °C for 30 s, 49 °C for 1 min, 72 °C for 1 min, and one cycle at 72 °C for 5 min. While for *cytK*, it was one cycle at 94 °C for 2 min, 35 cycles at 94 °C for 30 s, 52 °C for 1 min, 72 °C for 30 s, and one cycle at 72 °C for 10 min [40,41]. The *B. cereus* ATCC14579 strain was used as a positive control for amplifying the *gyrB, hbl*, and *plcR-cytK* genes. *B. cereus* BC133 strain was used as a positive control for *nhe* amplification. This strain was previously isolated and characterized in the laboratory from infant formula powdered [42].

### 2.4. Determination of B. cereus Biofilms

The production of biofilms was carried out with 1 mL of BHI broth in glass tubes (CLS99447161, Merck, Darmstadt, Germany) containing 50 μL of a 24 h culture. They were incubated for 48 h, and washed three times with phosphate-buffered saline or PBS 1X (NaCl 137 mM, KCl 2.7 mM, Na_2_HPO_4_ 10 mM, KH_2_PO_4_ 1.8 mM, pH 7.2); 1 mL of safranin was added and homogenized for 15 min, three washes with PBS solution were performed, and it was determined from the visual observation of a ring stained by the dye [42].

### 2.5. Determination of Genes Involved in the Formation of B. cereus Biofilms

The genes involved in biofilm formation were amplified by endpoint PCR: *sipW* (*sipW*-F AGATAATTAGCAACGCGATCTC, *sipW*-R AGAAATAGCGGAATAACCAAGC), *tasA* (*tasA*-F AGCAGCTTTAGTTGGTGGAG, *tasA*-R GTAACTTATCGCCTTGGAATTG), *calY* (*calY*-F AGGTATCGGGAGTT-CATCAG, *calY* R CAGCTTCTTGGTTGGCATTG), and *eps2* (*eps2*-F TGTTTTGAGCGGATTTGTTTTGT, *eps2*-R GATTGCTCTGCCAATGTCTTT), with the following reaction mix: 25 μL of REDTaq DNA Polymerase Ready Mix (Sigma-Aldrich, USA), 11 μL of sterile MiliQ water, 10 to 20 ng of genomic DNA and 0.02 μM of each oligonucleotide. For the *sipW*, *tasA*, and *calY* genes, the PCR protocol was one cycle at 94 °C for 5 min, 35 cycles at 94 °C for 30 s, 61 °C for 45 s, 72 °C for 45 s, and one cycle at 72 °C for 5 min [43,44]. For the *eps2* operon, the PCR protocol was one cycle at 95 °C for 5 min, 35 cycles at 94 °C for 30 s, 52 °C for 30 s, 72 °C for 90 s, and one cycle at 72 °C for 7 min. We used *B. cereus* ATCC14579 as a positive control [42].

### 2.6. Determination of Extracellular Enzymes and Cold Tolerance of B. cereus

Determination of the extracellular enzymes was performed by inoculation of 2 μL of a 24 h culture in different agar plates: 1% starch agar, 5% casein agar, 5% sheep blood agar, and MYP agar at 10% egg yolk emulsion. The agar plates were incubated at 30 °C for 24 h, and the 5% sheep blood agar at 4 °C for 10 days. After incubation, 30 μL of Lugol was added to demonstrate the amylolytic activity on 1% starch agar. For the proteolytic and lecithinase activity, the presence or absence of the hydrolysis halo was observed on 5% casein agar and 10% egg yolk emulsion. In the case of psychrotrophic capacity, the growth/non-growth at 4 °C of the strains was monitored for 10 days by visual observation, considering the day growth was observed as the detection time (DT).

### 2.7. Determination of Antimicrobial Resistance

The broth microdilution assay was used for each of the strains, using Mueller–Hinton (MH) broth, following the provisions of the CLSI M45:ED3 guide [45]. A total of 10 antibiotics (Oxoid, UK) were tested: ampicillin (0.12–16 μg/mL), ceftriaxone (2–64 μg/mL), ciprofloxacin (0.5–2 μg/mL), clindamycin (0.2–2 μg/mL), gentamicin (2–500 μg/mL), tetacycline (2–16 μg/mL), trimethoprim (0.5–9.5 μg/mL), vancomycin (1–64 μg/mL), kanamycin (2–500 μg/mL), and chloramphenicol (0.5–2 μg/mL). The inoculum of the *B. cereus s.l.* was prepared from a 24-h culture, first adjusting the culture concentration to a 0.5 McFarland turbidity standard (1 × 10^8^ CFU/mL). The final concentration of the inoculum in the microplate was 5 × 10^5^ CFU/mL. Each plate included a positive control (MH broth with inoculum without antibiotic) and a negative control (MH broth without inoculum with antibiotic). The microplates were incubated at 30 °C for 24 h. Ten μL of an MTT solution (3-(4,5-dimethylthiazol-2-yl)-2,5-diphenyltetrazolium bromide) was added to verify the growth and was incubated under dark conditions for 30 min at 30 °C. Formazan production evidenced the presence of viable cells.

### 2.8. Bioinformatics Analysis

To assess the evolutionary relationships between *Bacillus* strains, *16s rRNA* gene sequences from *B. cereus s.l.* and a reference strain were subjected to phylogenetic analysis using MEGA-X version 10.2.1 [39]. Seventeen sequences, including a reference sequence *B. cereus* ATCC 14893 (AJ310098.1) were used for phylogenetic analysis. Sequence data were subjected to two different phylogenetic reconstruction methods: Neighbor-Joining (NJ) and UPGMA. Positions containing missing data were removed from the dataset, genetic distances were estimated using the p-distance method, and a bootstrap analysis was carried out with 1000 resamples [39].

## 3. Results

### 3.1. Isolation and Identification of Bacterial Strains

Of the total samples analyzed, *B. cereus s.l.* was isolated and molecularly identified in 16 samples (12.9%, 16/124). Most samples were recovered from the air (62.5%, 10/16), while six were recovered from QF (37.5%, 6/16). Among the species identified in *B. cereus s.l.* by *16s rRNA* amplification and sequencing, *B. cereus sensu stricto* (*B. cereus*) was the most frequently isolated and identified species (81.25%, 13/16), followed by *B. wiedmannii* (12.5%, 2/16), and *B. manliponensis* (6.25%, 1/16). The 16 partial sequences of the *16s rRNA* gene were deposited in the GenBank database under accession numbers OQ507680-OQ507695 and are shown in Table 1.

### 3.2. Enterotoxigenic Profile of B. cereus

Regarding the toxigenic profile, most strains (93.75%, 15/16) presented at least one gene for some diarrheagenic toxins. Two strains of *B. cereus* and one of *B. wiedmannii* presented the genes for the three toxins analyzed (Nhe, Hbl, and CytK) (Figure 1).

Strains isolated from QF had at least one gene for diarrheagenic toxins. The strains studied did not contain the gene that codes for the emetic toxin (Table 2).

### 3.3. Determination of B. cereus Metabolic Traits

All the strains analyzed showed proteolytic activity; only three strains (18.75%, 3/16) were amylolytic. The strains produced lecithinase in different proportions (Table 3).

### 3.4. Determination of Biofilms and Gene Involved in the Biofilm Formation in Strains of B. cereus s.l.

All the strains were psychrotrophic. Most of the strains (87.5%, 14/16) formed biofilms (Table 4). With respect to the genes related to the production of biofilms, six strains (37.5%, 6/16) had both *tasA* and *sipW*, five strains (31.25%, 5/16) presented only *tasA*, 12 strains presented the *calY* gene (75%, 12/16), and the *eps2* operon was amplified in only five strains (31.25%, 5/16) (Figure 2) (Table 4).

### 3.5. Determination of Antimicrobial Resistance

Regarding antimicrobial resistance, all the *B. cereus s.l.* strains were resistant to penicillin, trimethoprim, and clindamycin, and 87.5% (14/16) were resistant to ceftriaxone. The strains were sensitive to ciprofloxacin, chloramphenicol, gentamicin, kanamycin, vancomycin, and tetracycline.

### 3.6. Phylogenetic Analysis

Phylogenetic analysis of *16s rRNA* allowed the identification of two different clusters. The air isolates of *B. cereus* were distributed in both clusters, but the cheese isolates were only located in only one cluster. A close phylogenetic relationship between isolates was found between the cheese isolates (B617 and B625) and the air isolates (B554 and B577). The *B. cereus* strains isolated from cheese (B617 and B625) had the same enterotoxigenic profile as one of the *B. cereus* strains isolated from the air (B554) (Figure 3).

## 4. Discussion

*B. cereus* is a microorganism associated with food poisoning due to the consumption of contaminated food [46]. Among the foods involved are dairy products in China [47]. Therefore, the presence of *B. cereus* from dairy farms has been analyzed in different studies in other countries [24,27,30,31,32,48]. In Mexico, *B. cereus* is currently not included in the sanitary legislation of any product. However, the circulation of this microorganism in different food matrices has been reported [49,50,51]. Regarding dairy products, the highest frequency has been reported in QF [25]. Therefore, in this study, the intentional search and molecular characterization of *B. cereus* was carried out in one dairy farm that produces fresh cheese in the same place. In this study, *B. cereus* was most frequently isolated from the air of the dairy farm, highlighting that this farm operates on earthen soil from milking to cheese production. In this sense, it has been described that *B. cereus* spores are commonly found in the air [24,48] and in farm soil [24,27,31,32,52], which is also confirmed in this study. Other studies have isolated it more frequently from feed, cow feces, and milk tanks [30,31,32]. A possible explanation may be related to the fact that, on the farm analyzed in this study, the cows are fed with grass that grows in the vicinity of the farm. In addition, the water for the cows is taken directly from the river that flows near the dairy farm, which may explain why it was not found in the feces.

Regarding the toxigenic profile, most of the strains presented at least one gene for one of the diarrheagenic toxins of *B. cereus*; the most common gene was the nonhemolytic toxin (Nhe); in this sense, the high frequency of genes of this toxin has been reported in strains isolated from the farm environment [27,30] and different dairy products [22,23,53,54,55], including those reported in the region. It is important to emphasize that the presence of the BL toxin gene has not been reported in the studies conducted in the region, including in the dairy industry [25,49,50,51]. However, in this study, the presence was found in strains isolated from farm air, which is consistent with what has been reported in dairy products in studies of other regions [22,23,53,54,55], and in farm environments [27,30,56]. In this study, the presence of the emetic toxin was not reported; it is important to highlight that other studies have described that emetic strains are not psychrophilic [57], which may explain the absence of strains with the presence of genes related to the emetic toxin.

It has been reported that *B. cereus* can form biofilms, which prevents its eradication from different food environments because the spores attached to the biofilm are protected from various disinfectants [58]. Therefore, it was important to determine the production of biofilms, and it was found that most of the strains produce biofilms on the glass; this property has been reported in strains isolated from ice cream [51]. Several operons have been described in biofilm formations that have multiple functions: the *eps2* operon has been related to bacterial adhesion to surfaces, cell-cell interaction, cell aggregation, and biofilms, while the *eps1* operon has been related to bacterial motility [44]. The *sipW-tasA-calY* operon is associated with the formation of amyloid-type fibers that are part of biofilms. SipW is a peptidase that processes TasA and CalY to their mature form and secretion [43]. CalY can be located at the cell surface, where it acts as an adhesin, thereby promoting the binding of bacterial cells to host tissues; in late stages and the biofilm, CalY is excreted into the extracellular medium and integrated into the matrix amyloid fibers, favoring TasA polymerization [59]. Most of the strains in this study presented at least one of these genes related to biofilm production, which is associated with the high percentage of biofilm production of these strains on glass. Cruz-Facundo et al. (2022) [50] reported a high frequency of the *sipW* and *tasA* genes and related this to the ability of the strains to produce biofilms in the shell of chicken eggs. It is worth mentioning that in addition to glass, the production of biofilms has been described in other materials and under different culture conditions [42,60], for which it is important to continue with the analysis of these strains.

Antibiotic therapy remains the principal treatment for *B. cereus* infections. However, the emergence of antibiotic-resistant *B. cereus* strains due to misuse of antibiotics or the acquisition of resistance genes leads to the failure of antibiotic treatment [61]. Therefore, in this study, the resistance to antibiotics was determined, and we found that all the strains of *B. cereus s.l.* were sensitive to vancomycin, kanamycin, gentamicin, tetracycline, ciprofloxacin, and chloramphenicol; on the other hand, they were resistant to ampicillin, ceftriaxone, clindamycin, and trimethoprim. Resistance to beta-lactams has been reported in strains isolated from different dairy products, as well as sensitivity to different groups of antibiotics, including glycopeptides, aminoglycosides, tetracyclines, quinolones, and phenicols [23,53,54,55].

Phylogenetic analysis of the *16s rRNA* allowed the identification of closely related strains circulating both in the cheese and the air of the dairy farm. Therefore, on this farm, the environmental presence of *B. cereus* could be the main route of cheese contamination. Therefore, new strategies related to the construction of spaces should be proposed to reduce its presence in the circulating air of this milk farm.

## 5. Conclusions

Strains of *B. cereus* were found in small-scale artisanal cheeses on a farm in southwestern Mexico, and airborne strains were also isolated from this farm. These strains possessed at least one of the most important enterotoxin genes of this microorganism. The strains exhibited metabolic characteristics that could favor the ability of these strains to break down dairy products, such as protease production and psychrophilic capacity. The *B. cereus* strains could produce biofilms, and the strains had at least one of the genes related to this process. The strains were resistant to beta-lactams and folate inhibitors. The contamination of the cheeses could be the circulating air in the QF production farm.

## Figures and Tables

**Figure 1 microorganisms-11-01290-f001:**
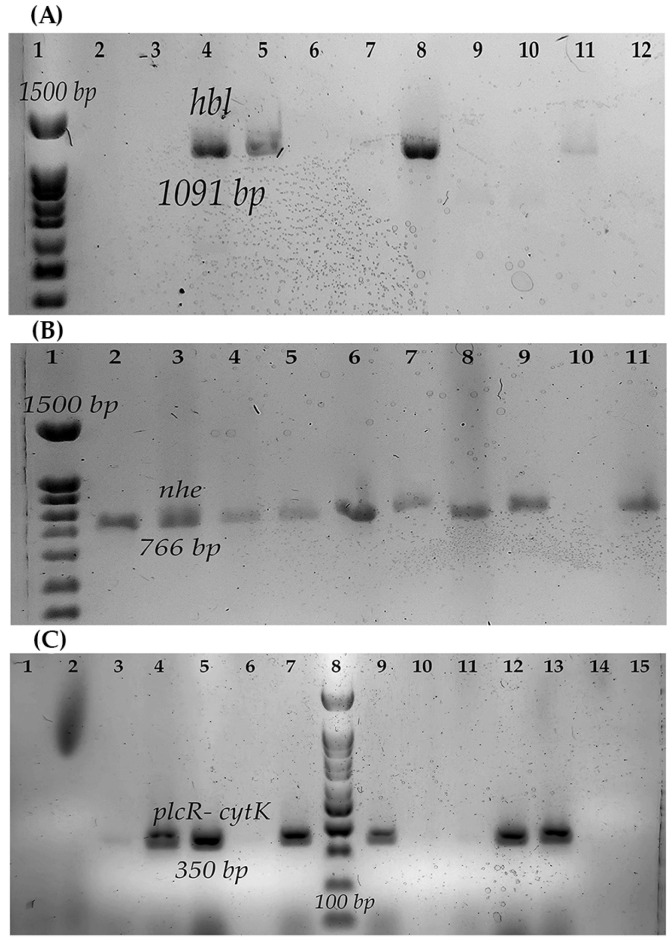
Toxins genes of the strain of *B. cereus s.l.* (**A**) *hbl* gene: 1, Molecular weight marker (100 bp); 2, B550; 3, B546; 4, B547; 5, B548; 6, B551; 7, B554; 8, Positive control *B. cereus* ATCC14579; 9, Negative control; 10, B565; 11, B576; 12, B578. (**B**) *nhe* gene: 1, Molecular weight marker (100 bp); 2, B546; 3, B547; 4, B548; 5, B550; 6, B551; 7, B552; 8, B554; 9, B565; 10, Negative control; 11, Positive control BC133. (**C**) *plcR- cytK* gene: 1, B546; 2, B565; 3, B576; 4, B547; 5, B548; 6, Negative control; 7, *B. cereus* ATCC14579; 8, Molecular weight marker (100 bp); 9, B550; 10, B619; 11, B623; 12, B551; 13, B552; 14, B577; 15, B578.

**Figure 2 microorganisms-11-01290-f002:**
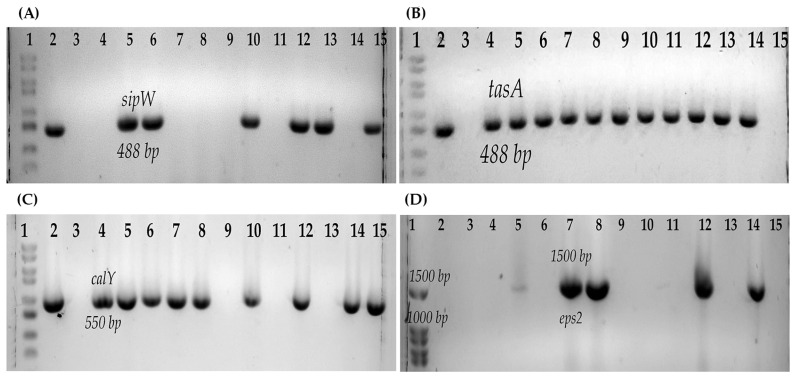
Genes associated with biofilm production of *B. cereus s.l.* (**A**) *sipW* gene: 1, Molecular weight marker (100 bp); 2, Positive control *B. cereus* ATCC14579; 3, Negative control; 4, B546; 5, B547; 6, B548; 7, B550; 8, B551; 9, B554; 10, B552; 11, B577; 12, B565; 13, B576; 14, B517; 15, B578. (**B**) *tasA* gene: 1, Molecular weight marker (100 bp); 2, Positive control *B. cereus* ATCC14579; 3, Negative control; 4, B547; 5, B548; 6, B552; 7, B554; 8, B565; 9, B576; 10, B578; 11, B617; 12, B619; 13, B622; 14, B625; 15, B623 (**C**) *calY* gene: 1, Molecular weight marker (100 bp); 2, Positive control *B. cereus* ATCC14579; 3, Negative control; 4, B546; 5, B547; 6, B548; 7, B550; 8, B551; 9, B617; 10, B552; 11, B619; 12, B554; 13, B623; 14, B565; 15, B576. (**D**) *eps2* operon: 1, Molecular weight marker (100 bp); 2, B546; 3, B547; 4, B548; 5, B551; 6, B565; 7, B550; 8, B552; 9, B576; 10, B577; 11, B617; 12, B578; 13, B619; 14, Positive control *B. cereus* ATCC14579; 15, Negative control.

**Figure 3 microorganisms-11-01290-f003:**
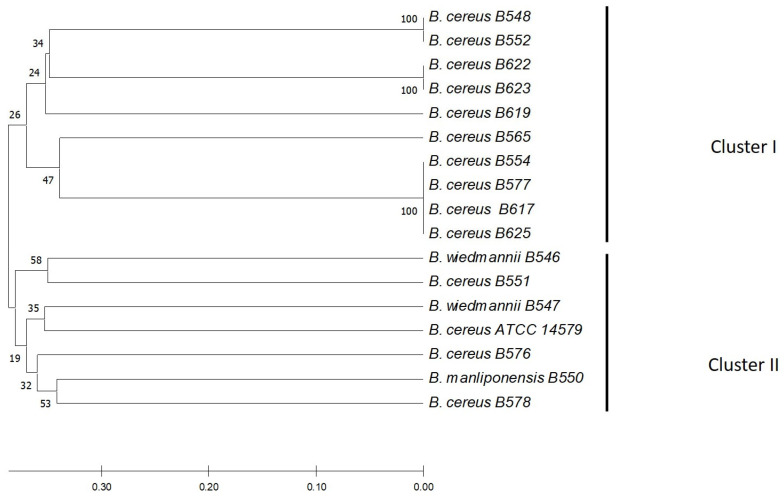
Neighbor-Joining (NJ) phylogenetic tree from sequencing the 16s *rRNA* gene of the strains isolated from the artisan fresh cheese production chain. The tree was constructed using the neighbor union method, and genetic distances were generated using the p-distance method. The percentage of replicate trees in which associated taxa clustered in the bootstrap test (1000 replicates) is shown next to the branches. The optimal tree is shown.

**Table 1 microorganisms-11-01290-t001:** Molecular identification of the strains of *B. cereus s.l.*

Strain	Accession Number on GenBank	Identification	Closest Related Bacteria	Similarity
B546	OQ507680	*B. wiedmannii*	*B. wiedmannii* FSL W8-0169	99.75
B547	OQ507681	*B. wiedmannii*	*B. wiedmannii* FSL W8-0169	99.88
B548	OQ507682	*B. cereus*	*B. cereus* ATCC 14579	100.00
B550	OQ507685	*B. manliponensis*	*B. manliponensis BL4-6*	89.55
B551	OQ507683	*B. cereus*	*B. cereus* ATCC 14579	100.00
B552	OQ507686	*B. cereus*	*B. cereus* ATCC 14579	99.69
B554	OQ507687	*B. cereus*	*B. cereus* ATCC 14579	99.16
B565	OQ507688	*B. cereus*	*B. cereus* ATCC 14579	99.86
B576	OQ507688	*B. cereus*	*B. cereus* ATCC 14579	99.87
B577	OQ507690	*B. cereus*	*B. cereus* ATCC 14579	100.00
B578	OQ507691	*B. cereus*	*B. cereus* ATCC 14579	99.87
B617	OQ507692	*B. cereus*	*B. cereus* ATCC 14579	100.00
B619	OQ507693	*B. cereus*	*B. cereus* ATCC 14579	100.00
B622	OQ507684	*B. cereus*	*B. cereus* ATCC 14579	99.86
B623	OQ507694	*B. cereus*	*B. cereus* ATCC 14579	100.00
B625	OQ507695	*B. cereus*	*B. cereus* ATCC 14579	100.00

**Table 2 microorganisms-11-01290-t002:** Toxin genes of the strain of *B. cereus s.l.*

Strain	Identification	Source	Diarrheagenic			Emetic
			*hbl*	*nhe*	*cytk*	*ces*
B546	*B. wiedmannii*	Air	−	+	−	−
B547	*B. wiedmannii*	Air	+	+	+	−
B548	*B. cereus*	Air	+	+	+	−
B550	*B. manliponensis*	Air	−	+	+	−
B551	*B. cereus*	Air	−	+	+	−
B552	*B. cereus*	Air	+	+	+	−
B554	*B. cereus*	Air	−	+	+	−
B565	*B. cereus*	Air	−	+	−	−
B576	*B. cereus*	Air	−	+	−	−
B577	*B. cereus*	Air	+	+	−	−
B578	*B. cereus*	Air	−	−	−	−
B617	*B. cereus*	QF ^1^	−	+	+	−
B619	*B. cereus*	QF	−	+	−	−
B622	*B. cereus*	QF	−	+	+	−
B623	*B. cereus*	QF	−	+	−	−
B625	*B. cereus*	QF	−	+	+	−

^1^ QF = Queso Fresco.

**Table 3 microorganisms-11-01290-t003:** Metabolic characteristics of strains of *B. cereus s.l.*

Strain	Lecithinase (mm)	Protease	Amylase
B546	0.47	+	−
B547	0.42	+	−
B548	0.55	+	+
B550	0.45	+	+
B551	0.50	+	−
B552	0.67	+	−
B554	0.73	+	−
B565	0.48	+	−
B576	0.55	+	−
B577	0.48	+	−
B578	0.52	+	−
B617	0.65	+	−
B619	0.60	+	−
B622	0.45	+	−
B623	0.57	+	−
B625	0.62	+	+

**Table 4 microorganisms-11-01290-t004:** Biofilms and genes involved in the biofilm formation of strains of *B. cereus s.l*.

Strain	Psychrotrophic Growth (DT)	Biofilm	*tasA*	*sipW*	*calY*	*eps2*
B546	1	−	−	−	+	−
B547	1	+	+	+	+	−
B548	1	+	+	+	+	−
B550	1	+	−	−	+	+
B551	1	++	−	−	+	−
B552	1	+	+	+	+	+
B554	1	+	+	−	+	+
B565	1	+	+	+	+	−
B576	1	++	+	+	+	−
B577	1	+	−	−	+	−
B578	1	+	+	+	+	+
B617	5	++	+	−	−	−
B619	5	+	+	−	−	−
B622	1	++	+	−	+	−
B623	5	−	−	−	−	−
B625	5	+	+	−	−	+

DT = detection time in days for visible growth.

## Data Availability

Data is contained within the article.

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
