# Peer review of "Bacillus cereus in the Artisanal Cheese Production Chain in Southwestern Mexico"

_microorganisms, 2023, doi:10.3390/microorganisms11051290_

Round 1

Reviewer 1 Report

B. cereus is a headache for the food industry due to its ubiquitous distribution, its ability to withstand harsh environmental conditions (due to its endospores and biofilm formation), and importantly, its ability to cause foodborne diseases and/or food spoilage. In this regard, monitoring foods for the presence of this pathogen is important to protect the public. The manuscript aims to isolate and characterize B. cereus strains from the artisanal cheese production chain in Mexico. Below, I have given some comments that I think would improve the quality of the manuscript.

 Major comments:

·         Line 92-96, the authors claim that they have collected 124 samples, but based on the detail, the sum of the samples they collected is about 130. The authors need to check/clarify this.

·         They have not mentioned the source /supplier of the antibiotics/kits they have used for susceptibility tests in the methods.

·         I wonder how they have adjusted the culture to 1x10^8 cfu/ml? Is it based on the OD-cfu relationship? They need to clarify this.

Figure 1.

·         Only a subset of isolates was analysed on the gel, especially in A and B. Any reason for that?

·         The labels on the gel images are not very visible. May increase the font sizes.

·         Gel C, lane 2, it is not clear whether there is a band or is just a smear. Is this positive or negative?

Table 3.  

·         Under the column “Psychrophile Day”, it is not clear to me what they are showing here. Is it the optimal temperature at which the strains grow?  If that is so, what does then Day refers to? As far as I have read in the methods, the authors have not presented such an analysis in the Methods section. I am not also entirely convinced whether these strains are psychrophilic or psychrotrophic. I would expect the authors to clarify this well.

Figure 2, it is good if they increase the font size of the labels.

In lines 312-315. Not clear, and needs a thorough revision.

Additional comments

It would also be good if they could cite some food poisoning data (if present) related to the consumption of such foods in Mexico or elsewhere.

Minor comments

·         Line 25, cereus should be italic

·         Line 27, is it only environmental samples? Do we consider cheeses as environmental as well?

·         Line 28, MYP agar

·         Line 31, antimicrobial resistance “test” or antimicrobial susceptibility test

·         Line 34, at least

·         Line 37, Air Isolates

·         Line 44, Bacillus, should be italic

·         Line 45, delete “it” is ubiquitous….

·         Line 49, check the grammar.

·         Line 63, replace the word isolation with monitoring or a related term

·         Lines 68-70, rewrite

·         Line 75, considered…

·         Line 80-81, cite a reference. What is “another” referring to here?

·         Line 86, it is not clear to me what the authors mean regarding “ although preliminary data are available at the point of sale”…

·         Line 99, I am not sure what “natural sinking” here

·         Line 123, Revise the grammar

·         Line 131, Subsequently,… S should be in small letter

·         Line 133, Revise the grammar

·         Line 135, toxin genes…

·         Line 152, cite a reference

·         Line 154, mention the supplier/catalog number of the glass tubes

·         156, define the ingredients of the PBS buffer including the pH, as there are many types of PBS.

·         156-158, it is not clear whether they assessed the biofilm formation visually or did a quantitative measurement (Eg. measuring absorbance)

·         Line 77, rewrite the sentences. Long and confusing!

·         Line 171, delete strain

·         Line 184, cite a reference

·         Line 198, Bacillus in italics

·         Table 1, Change the Closest related “bacterial” to “bacteria”

·         Line 222, two full stops

·         Line 228, two full stops

·         Line 230, may replace present with contain

·         Line 231, toxin genes…not toxins genes

·         Line 233, seems the title is not complete or needs a slight modification

·         Line 288, check English

·         Rewrite lines 302-305, not clear (or confusing).

·         Line 318, experimental relationship???

·         Line 327, check grammar;

·         Line 347, check English.

·         References: Please check the list for the proper use of italics and capitalization of the scientific names

I suggest the authors go through the manuscript and check the English, the commas, etc to avoid confusion. I would particularly suggest the authors revise the discussion part thoroughly. 

Author Response

We appreciate your comments regarding the methodology and the best visualization of the results. Also, we are grateful for your support on some English issues.

Reviewer 2 Report

This article presents a very detailed characterisation of Bacillus cereus present in the milk production and processing environment. The methodology is described in great detail, the results are presented clearly. In the discussion section of the results obtained, the threat posed by Bacillus ceresus in terms of food safety, in this case the fresh cheese, is indicated and discussed. 

 Only  minor editing of English language is required.

Reviewer 3 Report

The manuscript describes an interesting study on the presence of B. cereus in the production and environment of a farm that also produces its own artisan cheeses. B. cereus is a pathogen that deserves interest. The study has clear objectives and the results appear, in general, well discussed. However, beyond the observations detailed below, the study is not very original because, as the authors say in Discussion, there are several previous studies on the subject in Mexico. The particularity of this study is that it is studying a farm where QF cheese is also produced, but studying the presence of B. cereus in dairy products is not very new. The manuscript could be saved if some things are modified and several omissions are corrected in this version.

- The genus and species of microorganisms should always be written in italics. Check References, and lines 44 and 198 - line 80: another dairy products (or product?) - There are several problems related to what is reported about the sampling point and obtaining the samples: . southwestern (line 89) or northwestern (line 93) ? . I understand that 124 samples were collected during the month of August 2021. Information is missing about sampling points and frequency: a) what does "on the day" mean? 9 cheeses obtained the same day?; b) how often were all the other samples taken throughout that month?

- 2.6: Count of psychrotrophs: the plates were incubated at 4ºC. How long? How will a positive result be expressed? Days? days for what?

- Table 3: development at 4ºC is not a metabolic characteristic. Neither is biofilm formation. I suggest removing these data from the Table (the last 6 columns) and referring them elsewhere (another Table?)

- In Discussion, several concepts and data already referred to in Introduction are repeated. Adjust this

-

Author Response

We appreciate your comments regarding the manuscript. We hope that the methodological doubts will be resolved in the new manuscript. Regarding B. cereus and dairy products, the information is not further in Europe and Asia; However, in Latin America, the information is not enough. We believe that this study provides data to broaden this panorama. Because Latin America has too many tourist places, the consumption of regional artisan foods is common among residents and foreigners.

Round 2

Reviewer 1 Report

Most of the issues I have raised have been addressed in the first review. But there are a few that were not addressed properly. 

Major points

1. The description of the method that they have mentioned in line 181-183, and the data that they have presented in Table 4 is not compatible. In the methods, they mentioned that they have tested the growth of the strains at 4 degrees C, but there in Table 4, they have presented degrees, (Eg  1 degree C and 5 degrees). So they should clearly mention whether they have assessed the growth of the different isolates at different temperatures. Also, while they said in the methods that DT is Detection time, in the Table, the data under DT is presented in degree C. Therefore, it is still confusing, and the authors need to clarify this.

2. Figure 2. They have swapped the place of  Gel A and Gel B in the current version, perhaps by mistake, while the Legend is as it was. Please correct this.

3. Line 154, the authors just mentioned (Merck, Germany). However, the catalog number for this glass tube needs to be given. Merck perhaps produces many glass tubes and this should be specified, as this may affect the formation of biofilm.

Minor points

·         Line 68, B. cereus, italic

·         Line 133, delete “it”

·         Line 243, spelling mistake, psychrotroph

·         Line 248, biofilm ( B in capital letter)

·         Reference 33 and 53, scientific names are not italicized 

The quality of the manuscript will be improved if an English edit is also done. 

Reviewer 3 Report

The authors have responded well to almost all the observations made. Few issues remain to be adjusted:

- write in italics the genus and/or species of the microorganisms: in References 33 and 53

- in Reference 35 "Shiga" should not be written in italics

-2.6: according to the 10 days but it must also be indicated in 2.6 because it is part of the method

- Table 4: indicate what "DT" is

                  Change "psychrotrophic bacteria" to "psychrotrophic growth" because what is indicated is whether the bacteria developed or not. Without these clarifications, it is not understood what indicates "1o "

- Leave the Discussion like this because I don't understand how it could start on line 303 of the new version....
